# The Free Energy of Nucleosomal DNA Based on the Landau Model and Topology

**DOI:** 10.3390/biom13121686

**Published:** 2023-11-23

**Authors:** Huimin Yang, Xuguang Shi

**Affiliations:** College of Science, Beijing Forestry University, Beijing 100083, China; yanghm@bjfu.edu.cn

**Keywords:** nucleosome core particle, DNA free energy, minimum free energy principle, Landau theory, Hopfion model

## Abstract

The free energy of nucleosomal DNA plays a key role in the formation of nucleosomes in eukaryotes. Some work on the free energy of nucleosomal DNA have been carried out in experiments. However, the relationships between the free energy of nucleosomal DNA and its conformation, especially its topology, remain unclear in theory. By combining the Landau theory, the Hopfion model and experimental data, we find that the free energy of nucleosomal DNA is at the lower level. With the help of the energy minimum principle, we conclude that nucleosomal DNA stays in a stable state. Moreover, we discover that small perturbations on nucleosomal DNA have little effect on its free energy. This implies that nucleosomal DNA has a certain redundancy in order to stay stable. This explains why nucleosomal DNA will not change significantly due to small perturbations.

## 1. Introduction

It is well known that DNA is genetic material. DNA is a semi-flexible polymeric macromolecule [1,2,3]. The diameter of the cross-section of a double helix is about 20 Å [3]. But its length is up to the centimeter scale [4,5,6]. DNA is prone to forming disordered knots, which can have detrimental effects on its biological functions. In order to avoid this entanglement, eukaryotes have evolved nucleosomes, which wrap the DNA around the histone octamer, to achieve an unentangled compaction of DNA. Then, the nucleosome forms chromatin fiber. The chromatin fiber further folds into topologically associated domains via the process of loop extrusion mediated by chromosome structure maintenance complexes [7]. Finally, DNA is compacted into a very small volume of the cell nucleus and performs various biological functions correctly. The nucleosome is the basic unit of chromatin [8]. There are four types of proteins to form the histone octamer. The histone octamer is wrapped around by approximately 145–147 bp in a left-handed superhelix. The turns of the wrapping are about 1.7 [9,10,11]. All of these constitute complex structures called nucleosome core particles (NCPs). The DNA on the histone octamer is called nucleosomal DNA. The H1 histone associates with the linker DNA on the entry and exit sites of the NCPs [12]. The nucleosomal DNA is locked at the entry and exit points of the NCPs by histone H1. Thus, we take nucleosomal DNA as a closed circular DNA, which can be considered a knot. Moreover, nucleosomes are a dynamic platform for connecting and integrating various biological processes [13,14,15]. Therefore, it is necessary to reveal the dynamic changes in the nucleosome structure.

In recent years, a lot of work has been carried out regarding the nucleosome DNA’s conformation and free energy [13,16,17,18,19,20]. With the development of technology, more and more single-molecule mechanics experiments have shown that force and free energy are important parameters for understanding the biochemical processes at the molecular scale [21,22,23,24,25]. Free energy is also the bridge between theoretical calculations and experimental data [26]. Especially, in Ref. [27], the authors define the fraction of native contacts Q to represent the free energy of the histone octamer. They obtained the free energy landscape during the process of the assembly of the histone octamer in the frame of a predictive coarse-grained DNA–protein model. Then, the free energy profile of the full nucleosome disassembly as a function of the radius of gyration is presented. The results show the number of base pairs bound to the histone octamer decreases with an increase in the radius of gyration. The free energy profile has an inflection point and there is a minimal value near the radius $R_{DNA}=45\;\overset{\circ }{A}$. They also verified the asymmetry of DNA unwrapping. How does the free energy of the nucleosomal DNA wrapped around the histone octamer behave?

To answer the above question, we take closed circular DNA as a knot, which is described by the linking number. Then, we combine the Landau theory and the Hopfion model to derive the free energy density of the DNA. The free energy density has a relationship with the linking number and other geometrical parameters [28]. Finally, we apply the free energy density to the nucleosomal DNA and discuss how the linking number and geometrical parameters affect the free energy. The results show the free energy of the nucleosomal DNA is at a lower level. The formation of a stable nucleosomal DNA may not be accidental and may follow the energy minimum principle. In addition, we also find that the nucleosomal DNA has a certain redundancy from the viewpoint of free energy. The small changes in the conformation of the nucleosomal DNA have little effect on its free energy. These ensure to keep the nucleosomal DNA relatively stable. When the perturbations are removed, DNA can automatically revert to its original structure.

## 2. Theory

### 2.1. The Topology and Mechanical Properties of Nucleosomal DNA

DNA undergoes bending and twisting to form various complex topological structures; its topology properties follow White’s formula: $SL=W_{r}+T_{w}$ [29]. SL is the self-linking number, which is the number of strands crossing the other when double-stranded DNA is completely unwound. It is an integer that does not change as long as the double strands do not break. The self-linking number can be calculated from a two-dimensional diagram of a framed curve, obtained using its projection on a selected plane. Then, the self-linking number is given by $L_{k}=W_{r}+T_{w}$. $W_{r}$ is the writhing number of the DNA helix axis and $T_{w}$ is the twisting number of the DNA. Both $W_{r}$ and $T_{w}$ can be non-integers, which are not topological numbers. Canonical NCPs are generally formed of B-DNA with 145–147 bp, which wraps around a histone octamer approximately 1.7 times in a left-handed helix. For example, 1KX5 [30] and 5AVC [31] are the NCPs of humans, which are selected from the Protein Data Bank. Their structures are shown in Figure 1a–d. The schematic diagram of the NCPs is displayed in Figure 1e,f.

In this paper, we assume the nucleosomal DNA is a B-form double helix with 147 bp. Therefore, the linking number of the nucleosomal DNA can be given by [3]
(1)$$L_{k}=\frac{N}{n_{i}},$$
where N is the total number of base pairs of the nucleosomal DNA and $n_{i}$ is the number of the base pairs per turn of the DNA double helix, where $n_{i}=10.5$ [3,10,11]. Therefore, the linking number of the nucleosomal DNA with 147 bp is $L_{k}=\frac{147}{10.5}=14$. The nucleosomal DNA wraps around a histone octamer by 1.75 turns in a left-handed fashion. So, Wr=−1.75. Based on White’s formula, it is known that the writhing number can transfer to the twisting number for closed circular DNA. The twisting number changes with $-\Delta L_{k}$ when the writhing number changes with $\Delta L_{k}$. This leads the linking number $L_{k}$ to be a topological invariant if the closed circular DNA does not break. For a linear chain, the linking number is not a topological invariant. Therefore, the twisting number of the nucleosomal DNA is Tw=14−(−1.75)=15.75.

It is well known that the diameter of a B-DNA double helix is d=2 nm and the distance between the adjacent base pairs is l=0.34 nm. The radius of the NCPs is r=5.5 nm [32,33]. The sketch of the top view of the NCPs as shown in Figure 1f. From the top view, it can be seen that the nucleosomal DNA helix axis forms a circle. The center of the circle is exactly the geometric center of the NCPs. The distance between the DNA helix axis and the geometrical center of the NCPs is the radius of the circle. Considering the diameter of the NCPs is 11 nm and the diameter of the B-DNA is 2 nm, the radius of the circle is $\tilde{r}=4.5\;\mathrm{nm}$. The bending is a key factor in the DNA conformation, which has a significant effect on the free energy of DNA. So, we should define a quantity to describe it. Taking one pitch of the nucleosomal DNA as a unit of arc length, the average bending angle corresponding to the unit arc length is $\theta =\frac{\left|W_{r}\right|\times 360{}^{\circ}}{T_{w}}=\frac{1.75\times 360{}^{\circ}}{15.75}=40{}^{\circ}$. This angle can be used to describe the bending of the helix.

The mechanical properties of the nucleosomal DNA can be expressed using the bending rigidity and torsional rigidity. Let us consider the bending persistence of the B-DNA is $l_{p}=B/k_{B}T$ [1], where $l_{p}=50\;\mathrm{nm}$. B is the bending rigidity of the B-DNA. $k_{B}$ is the Boltzmann constant, $k_{B}=1.38\times 10^{-23}\;J/K$ [34,35]. T is the absolute temperature, where T=300 K. Therefore, the bending rigidity of the B-DNA is taken as $B=2.07\times 10^{-19}\;J\cdot \mathrm{nm}$. Similarly, the torsional rigidity of the B-DNA is $C=3.933\times 10^{-19}\;J\cdot \mathrm{nm}$, where its torsional persistence length of $l_{t}=95\;\mathrm{nm}$ [36].

### 2.2. The DNA Free Energy Relating to Its Topology

In Ref. [37], one end of the linear DNA is fixed on the glass coverslips and the other end is fixed on the magnetic beads. The magnetic tweezers formed using an external magnetic field can control the magnetic beads. They applied a negative torque to the DNA by rotating the magnetic tweezer. Then, they found that the minute negative superhelicity can facilitate the B–Z transition at low tension with the help of the single-molecule Förster resonance energy transfer (FRET). The experimental schematic is shown in Figure 2a. If $\tau >\tau_{c}$, the DNA left-handed helical DNA converts into the right-handed helical DNA. If $\tau <\tau_{c}$, the opposite is true. In this process, the local interactions between the successive base pairs play an important role in the transition. This local interaction can be described using the chiral gauge potential ρ, which is just the rotational angle between the successive base pairs. ρ>0 and ρ<0 represent the DNA in the right-handed helix and left-handed helix, respectively. Thus, the effective potential energy of the DNA [38] is expressed as:(2)$$V_{eff}(\rho )=V_{0}{\left[{\left(\frac{\rho }{\omega_{0}}\right)}^{2}-1\right]}^{2}-\left(\tau -\tau_{c}\right)\rho ,$$
where $V_{0}=6\times 10^{-21}\;J$. $\omega_{{}_{0}}$ is the helix rotation angle between the consecutive base pairs; $\omega_{{}_{0}}=2\pi /n_{{}_{i}}=0.6\;\mathrm{rad}/\mathrm{bp}$ [36]. τ is the external torque applied to the ends of the linear DNA molecule. The critical torque $\tau_{c}$ is $-7.9\times 10^{-21}\;J$. The effective potential energy diagram is shown in Figure 2b. Therefore, the mutual transformation between the different conformations of DNA can be taken as the phase transition. Then, the chiral gauge potential is the bridge between the local interaction of the DNA base pairs and global conformation.

The Landau theory is the basic theory for describing the phase transition. It introduces symmetry breaking into the phase transition and expands the free energy density function with the power series of the order parameters. The equilibrium stability of the structure can be determined using the minimum of the free energy density function [40]. In the frame of the Landau thoery, the free energy density relating to the conformation of the DNA can be given as [38],
(3)$$F=\frac{k_{1}}{2}\frac{d\Psi^{*}}{ds}\frac{d\Psi }{ds}+\frac{k_{2}}{2}{\left|\Psi^{*}\frac{d\Psi }{ds}\right|}^{2},$$
where $k_{1}=4B$, and $k_{1}+k_{2}=4C$. $k_{1}$ and $k_{2}$ are physical quantities relating to the DNA bending and torsion, respectively. s is the arc length parameter along the DNA helix axis. Ψ is the two-component order parameter that describes the conformation of the DNA. The relationship between the two-component order parameter and the Euler angle is given as:(4)$$\Psi =(\cos\frac{\theta }{2}e^{-i\frac{\left(\phi +\chi \right)}{2}},\sin\frac{\theta }{2}e^{i\frac{\left(\phi -\chi \right)}{2}}),$$
where ϕ, θ and χ are the Euler angles. The first part of Equation (4) describes the conformation of the DNA, or the superhelix. The second part describes the internal interactions between the base pairs, which represents the double helix. The two-component order parameter can be obtained by operating a Euler rotation on the state
(5)$$\left(\begin{array}{c} 1\\ 0 \end{array}\right).$$
The Euler rotation is given as:(6)$$T=\left(\begin{array}{cc} \cos\frac{\theta }{2}e^{-i\frac{\phi +\chi }{2}} & \sin\frac{\theta }{2}e^{-i\frac{\phi -\chi }{2}}\\ \sin\frac{\theta }{2}e^{i\frac{\phi -\chi }{2}} & \cos\frac{\theta }{2}e^{i\frac{\phi +\chi }{2}} \end{array}\right).$$
The steps of the Euler rotation are shown in Figure 3. Substituting Equation (4) into Equation (3), we have:(7)$$F=\frac{B}{2}\left[{\left(\frac{d\theta }{ds}\right)}^{2}+{\left(\frac{d\phi }{ds}\right)}^{2}\sin^{2}\theta \right]+\frac{C}{2}{\left(\frac{d\phi }{ds}\cos\theta +\frac{d\chi }{ds}\right)}^{2},$$
where $\frac{d\phi }{ds}$, $\frac{d\theta }{ds}$ and $\frac{d\chi }{ds}$ represent the rotation degree of the DNA helix axis, the bending degree of the DNA helix axis and its deviation degree from the plane perpendicular to the helix axis, respectively. The first term in Equation (7) is the curvature of the DNA helix. The second term is the torsion of the DNA helix [26], which is $\tau =\frac{d\phi }{ds}\cos\theta +\frac{d\chi }{ds}$. We can obtain the twisting number of DNA by integrating it: $T_{w}=\frac{1}{2\pi }\int_{0}^{L}\tau ds$ [41], where L is the length of the DNA. If we know the curvature and torsion of the helix, Equation (7) also can be used to investigate the free energy of the nucleosomal DNA when it is decomposed from the histone octamer.

Linear DNA is the major subject of in vitro experiments. However, DNA rarely exists in a simple linear configuration in vivo [42]. In eukaryotes, DNA wraps around the histone octamer to form nucleosome, and via the process of loop extrusion, chromatin fibers composed of nucleosomes can be further folded into chromosome structures, mediated by chromosomal structure maintenance complexes [7]. The loops formed during the process of loop extrusion can be represented and modeled as closed circles [43,44]. In addition, the closed circular DNA of prokaryotes also twists into superhelical structures [45,46,47]. The nucleosomal DNA can be modeled as closed circular DNA because it is locked by the histone H1, the topological object is called a DNA knot. The topology of the DNA knot follows White’s formula. In addition, the torus knots can be described by Hopfions, which are closed twisted baby Skyrmion strings. In Ref. [48], the two-component order parameters relating to the torus knots are introduced as:(8)Ψ=(cosRe−iPφ)sinReiQΘ),
where φ is the rotational angle of the base pairs in the toroidal direction. Θ is the rotational angle in the poloidal direction. They are shown in Figure 4. P is the twisted number of the knot in the toroidal direction. Q is the twisted number of the knot in the poloidal direction. The Hopfions can be described using the Hopf charge, which is defined as,
(9)W=PQ,
here, the Hopf charge is just the linking number of the nucleosomal DNA, that is, $W=L_{k}$.

Based on Equations (4) and (8), if the DNA knot can be expressed as a torus knot or a Hopfion, we have the following relationship
(10)$$\begin{array}{l} \theta =2R\\ \phi =P\varphi +Q\Theta \\ \chi =P\varphi -Q\Theta . \end{array}$$
These equations are just the mappings that build the relationships between the DNA knots and torus knots. In the case of a nucleosome, the NCPs can be bended into the torus under the map (10). Then, the DNA knot becomes the torus knot. In fact, we assume the number P is just the writhing number $W_{r}$. R is the bending angle of the DNA helix axis after the mapping. Based on the Equation (10), the value range of R is $\left(0,\frac{\pi }{2}\right)$. As shown in Figure 4, these angles are not real angles, but they relate to the Euler angles. This map also gives us a bridge to find the relationships between the linking number and Hopf charge. Although the relations between the general knots and torus knots are not very clear [48,49,50], Equation (10) provides us with a new perspective to explore the DNA knot. We can substitute Equation (10) into Equation (7),
(11)$$\begin{array}{l} F=\frac{B}{2}\left[4{\left(\frac{dR}{ds}\right)}^{2}+\left(P^{2}{\left(\frac{d\varphi }{ds}\right)}^{2}+Q^{2}{\left(\frac{d\Theta }{ds}\right)}^{2}+2PQ\frac{d\varphi }{ds}\frac{d\Theta }{ds}\right)\sin^{2}(2R)\right]\\ +\frac{C}{2}\left[P^{2}{\left(\frac{d\varphi }{ds}\right)}^{2}4\cos^{4}R+Q^{2}{\left(\frac{d\Theta }{ds}\right)}^{2}4\sin^{4}R-2PQ\frac{d\varphi }{ds}\frac{d\Theta }{ds}\sin^{2}(2R)\right], \end{array}$$
where $\frac{d\varphi }{ds}$ is the change rate of the angle φ. $\frac{d\Theta }{ds}$ is the change rate of the angle Θ. $\frac{dR}{ds}$ is the change rate of the bending angle of R. In this paper, we assume $\frac{dR}{ds}=0$. This means the bending angle R of DNA does not change. Supposing $\frac{d\varphi }{ds}=m$, $\frac{d\Theta }{ds}=n$. Finally, we can obtain:(12)$$F=\frac{B}{2}\left[{\left(Pm+\left(\frac{W}{P}\right)n\right)}^{2}\sin^{2}(2R)\right]+2C{\left[Pm\cos^{2}R+\left(\frac{W}{P}\right)n\sin^{2}R\right]}^{2}.$$
If the length of the DNA is b, its free energy is:(13)$$E=\int_{0}^{b}Fds.$$

## 3. Results

### 3.1. The Free Energy of Nucleosomal DNA

Now, we apply the free energy to the nucleosomal DNA in the NCPs. Based on the previous discussion, the linking number $L_{k}$ of the nucleosomal DNA with 147 bp is $L_{k}=14$. The average bending angle of the nucleosomal DNA per unit arc length on the histone octamer core is θ=40°. By recalling $L_{k}=W$ and θ=2R, we have W=14 and R=20° after mapping the nucleosomal DNA onto the toroidal domain wall. The schematic diagram for the NCPs is given in Figure 5a. Since P and Q are integers, there are four possibilities for their values: P=1 and Q=14; P=2 and Q=7; P=7 and Q=2; P=14 and Q=1, respectively.

Let us assume that the change rates of φ and Θ are identical: m=1 and n=1. The free energy profiles and the histogram of the nucleosomal DNA with P=1,2,7,14 are shown in Figure 5b. We find that the free energy of the nucleosomal DNA has a minimum when P=2. According to general experimental data relating to nucleosomes R=20°, the free energy of the nucleosomal DNA with P=1.75 is very close to the minimum. This implies the free energy of nucleosomal DNA may follow the minimum energy principle in order to stay stable. The construction of the NCPs may not be randomly generated, but carefully designed. The free energy profiles with different R values are plotted in Figure 5c. These profiles show that the minimum of the free energy depends on the value of R. For example, the minimum of the free energy is about $1071\cdot bk_{B}T$ at P=2.1 when R=22°. The free energy profile is plotted as the blue line in Figure 5c. The minimum of the free energy is about is $938\cdot bk_{B}T$ at P=1.9 when R=18°. The free energy profile is plotted as the black line in Figure 5c. The minimum of the free energy also increases with R and decreases when R decreases. We also find the value of P corresponding to the minimum of the free energy increases when R increases. We present a histogram that shows the minimums of the free energy with different R values in Figure 5d. We also find the differences between the minimums with different R values are very small.

### 3.2. The Free Energy of Nucleosomal DNA with Different Diameters of Histone Octamers

The diameter of the histone octamer may change in the processes of replication, transcription and so on. These will lead to the changes in the parameters W and R. In the following, we discussed how the parameters W and R affect the DNA’s free energy.

In order to explore how R affects the free energy of nucleosomal DNA, we set m=1, n=1 and W=14. In this case, the diameter of the histone octamer core is 7 nm, which is used as a benchmark. We also consider the length and pitch of DNA are invariant. If the eight histones are so loosely bound, the radius of the histone octamer increases. The bending angle corresponding to the unit arc length of the nucleosomal DNA becomes smaller. The turns of the nucleosomal DNA wrapping around the histone octamer core will decrease. The result is $W_{r}$ becomes smaller, as shown in Figure 6a. To an extreme extent, the nucleosomal DNA barely bends anymore and becomes almost straight-stranded when the eight histones are disassembled, as shown in Figure 6b. Conversely, if the eight histones are more tightly bound, the radius of the histone octamer core decreases. This leads to the nucleosomal DNA bending at a large angle. Then, the turns of the nucleosomal DNA wrapping around the histone octamer core will increase, that is, $W_{r}$ will become larger. This is shown in Figure 6c. The free energy profiles of the nucleosomal DNA are plotted in Figure 6d. It presents the minimum of the free energy increases as the angle R increases. The value of P corresponding to the minimum of the free energy also increases. This indicates that DNA needs more energy to wrap around the histone octamer core tightly. In turn, the tightness of the binding of the eight histones also affects the free energy of the nucleosomal DNA. We present the free energies at P=2 and P=1.75 when R=5°,12°,20°,30° in Figure 6e. We find the differences between the free energies are small, especially for R=5°,12°,20°. These results mean small changes in R have little effect on the free energy of the nucleosomal DNA. These show the conformation of nucleosomal DNA has some redundancy. That is, a small change in R slightly affects the stability of the nucleosomal DNA.

### 3.3. The Free Energy of Nucleosomal DNA with Different Linking Numbers

In this section, we discuss how the linking number Lk affects the free energy of nucleosomal DNA. We set m=1, n=1, R=20°. Let us assume the diameter of the histone octamer core and pitch of the nucleosomal DNA wrapped around the histone octamer are constant. When the DNA length is 63 bp, the number of turns of the nucleosomal DNA wrapping around the histone octamer core is 0.75. Then, its linking number is 6. A sketch of its structure is shown in Figure 7a. The free energy profile is plotted as the orange line in Figure 7b. When the DNA length is 168 bp, the number of turns of the nucleosomal DNA wrapping around the histone octamer core is 2. Then, its linking number is 16. The sketch of its structure is shown in Figure 7c. The free energy profile is plotted as the red line in Figure 7b. When the DNA length is 189 bp, the number of turns of the nucleosomal DNA wrapping around the histone octamer core is 2.25 and its linking number is 18. The sketch of its structure is shown in Figure 7d. The free energy profile is plotted as the black line in Figure 7b.

Comparing the free energy profiles in Figure 7b, it can be found that the minimum of the free energy becomes larger as the linking number increases. The P value corresponding to the minimum free energy also increases. This is consistent with the fact that the longer the nucleosomal DNA is, the more energy it has. The free energy of the nucleosomal DNA is low when the nucleosomal DNA is too short. The number of turns of the nucleosomal DNA wrapping around the histone octamer is even less than one. This makes such structure unstable and meaningless, as shown in Figure 7a. When the nucleosomal DNA is too long, its free energy becomes big. This leads the NCPs to be unstable. On the other hand, the height of the histone octamer core should be less than 6 nm. The diameter of the nucleosomal DNA is 2 nm and its superhelical pitch is 2.8 nm [33]. So, this means that the number of the turns of the nucleosomal DNA wrapping around the histone octamer core is not greater than 3. Therefore, the DNA cannot be too long. The sketch shown in Figure 7d would not be possible in organism.

The histogram of the free energy of nucleosomal DNA with different W values is presented at P=2, P=1.75 when $L_{k}=6,14,16,18$ in Figure 7e. We find the difference between these free energies is small except for $L_{k}=18$. That means the free energy of the nucleosomal DNA only changes slightly as long as the length and conformation of the nucleosomal DNA do not change dramatically. The NCPs can stay in a dynamic stable state. This reveals that the nucleosomal DNA on the histone octamer has some redundancy, which keeps its conformation stable.

## 4. Discussion

The parameters of nucleosomal DNA, such as the bending angle θ, the bending elastic rigidity B and torsion elastic rigidity C, can be calculated and found from the general experimental data. In this paper, we have θ=20°, $B=2.07\times 10^{-19}\;J\cdot \mathrm{nm}$ and $C=3.933\times 10^{-19}\;J\cdot \mathrm{nm}$. Some other parameters, such as the change rate of the toroidal angel m and the change rate of the poloidal angle n on the torus, are not easy to calculate. In general, we set m=n=1. We find there is a minimum in the free energy profile of the nucleosomal DNA under any conditions. The structure of the nucleosomal DNA should be the most stable when its free energy takes the minimum according to the energy minimum principle. In this model, the free energy of real nucleosomal DNA takes a minimum at P=2. We find the free energy of the nucleosomal DNA with P=1.75 is very close to the minimum. Therefore, we believe the construction of NCPs is not randomly generated, but has been carefully designed. The formation of NCPs may follow the energy minimum principle.

Furthermore, it can be found that the changes in the free energy are very gentle near the minimum when the parameters vary slightly based on Figure 5c, Figure 6b and Figure 7b. This means that small perturbations have little effect on the structure and free energy of the nucleosomal DNA. We infer the nucleosomal DNA structure should be in a dynamically stable state near the minimum of free energy. When perturbation is removed, we think that nucleosomal DNA should automatically return to a stable state. This indicates that nucleosomal DNA has some redundancy. The ability of nucleosomes to serve as dynamic platforms for connecting and integrating multiple biological processes may dependent on their redundancy. When nucleosomal DNA is involved in biological processes, such as transcription and replication, the nucleosomal DNA needs to be unwrapped from the histone octamer core. If the degree of unwrapping is small, the unwrapped DNA can be quickly rewrapped. However, when the external environment changes significantly, resulting in huge changes in the structural parameters, the corresponding free energy will also increase dramatically. At this time, the structure of nucleosomal DNA may be disrupted and it is difficult to restore it to its original configuration. These results are consistent with the phenomena observed in the previous studies [17,19,51]. This also may be one of the reasons why the lengths of nucleosomal DNA in different species are different, but the number of turns wrapping around the histone octamer core is always around 1.7. Moreover, we hypothesize that DNA also could fold in the same way without the histone octamer because this is determined by the mechanical properties of the DNA. However, the histone octamer is the skeleton that keeps the nucleosome more stable. We also think that other proteins, such as bacterial Fis, HU and others, also can fold to form complex geometrical conformations. These conformations are also determined by their mechanical properties.

## Figures and Tables

**Figure 1 biomolecules-13-01686-f001:**
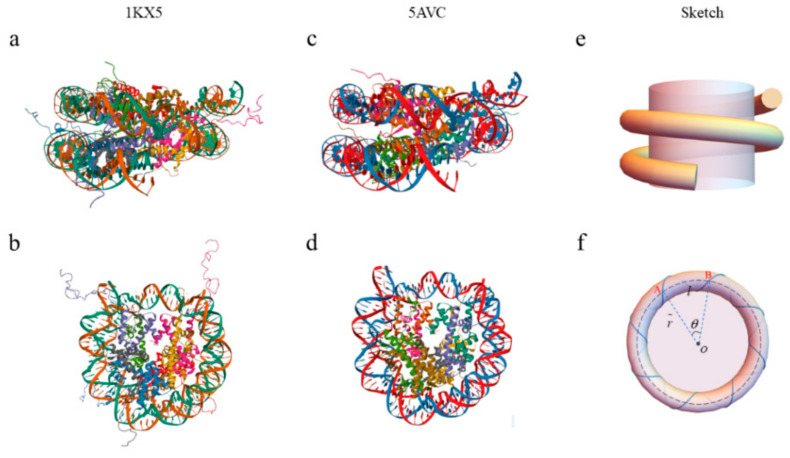
Structures of the NCPs. (**a**) The front view of 1KX5. (**b**) The top view of 1KX5. (**c**) The front view of 5VAC. (**d**) The top view of 5VAC. (**e**) Sketch of the front view of the NCPs. (**f**) Sketch of the top view of the NCPs. Where the gray part represents the histone octamer core, the pink/purple part represents the DNA double helix. The black dotted line represents the axis of the DNA double-helix. A and B represent the ends of one turn of a helix of the nucleosomal DNA. O is the geometry center of the NCPs. θ is the angle corresponding to the arc length joining the ends of one turn of the helix of the nucleosomal DNA. l is the pitch of the nucleosomal DNA. $\tilde{r}$ is the distance from the geometric center of NCPs to the axis of DNA double-helix.

**Figure 2 biomolecules-13-01686-f002:**
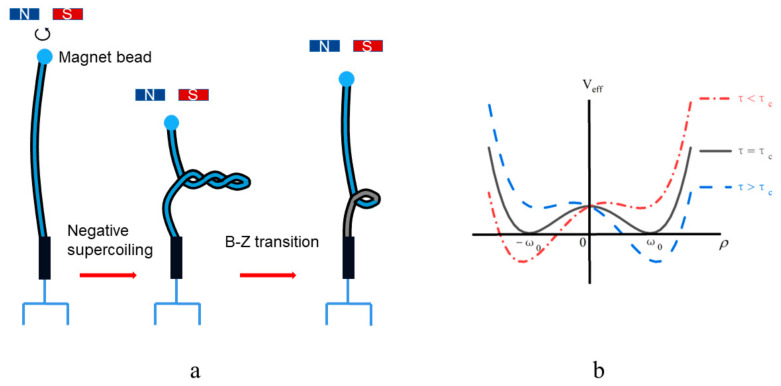
(**a**) Schematic diagram of the magnetic tweezer experiment [37,38,39]. (**b**) The effective potential energy profile.

**Figure 3 biomolecules-13-01686-f003:**
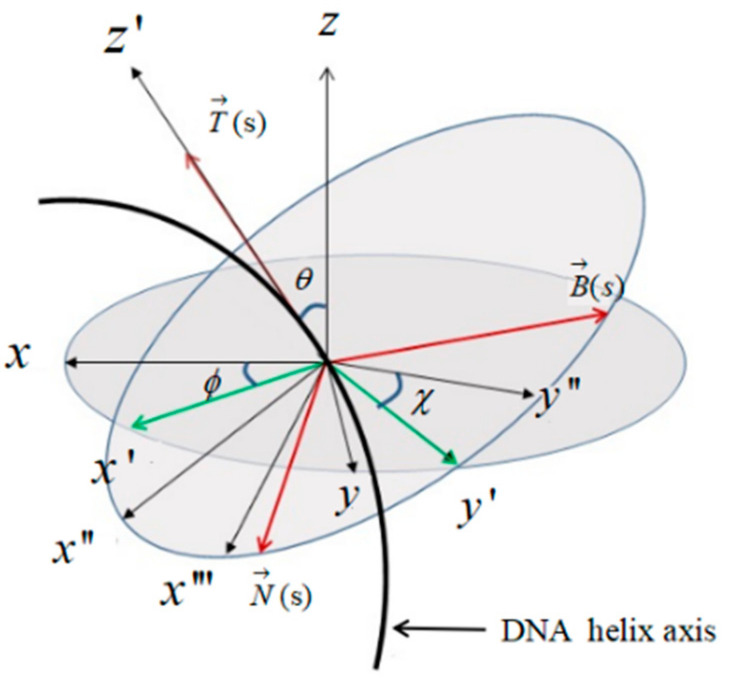
Euler rotation of the DNA helix axis. Firstly, the DNA helix axis is rotated clockwise around the z axis by angle ϕ. Then, the helix is rotated around $y^{\prime }$ by angle θ. Finally, the helix rotates around the $z^{\prime \prime }$ axis by angle χ. The value range of θ is (0,π).

**Figure 4 biomolecules-13-01686-f004:**
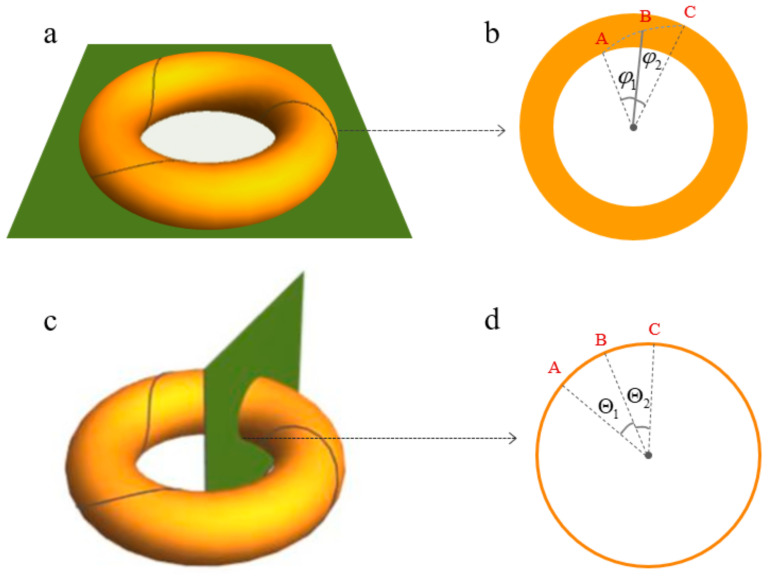
Toroidal domain wall. Where W=3, (P,Q=3,1). (**a**) Schematic diagram of the transverse section of the torus. (**b**) Schematic diagram of the angle corresponding to DNA per unit arc length in the transverse section of the torus. A, B and C are the points at which three successive sites of DNA are projected onto the circle in the transverse section. (**c**) Schematic diagram of the poloidal section of the torus. (**d**) Schematic diagram of the angle corresponding to DNA per unit arc length in the toroidal section of the torus. A, B and C are the points at which three successive sites of DNA are projected onto the circle in the poloidal section.

**Figure 5 biomolecules-13-01686-f005:**
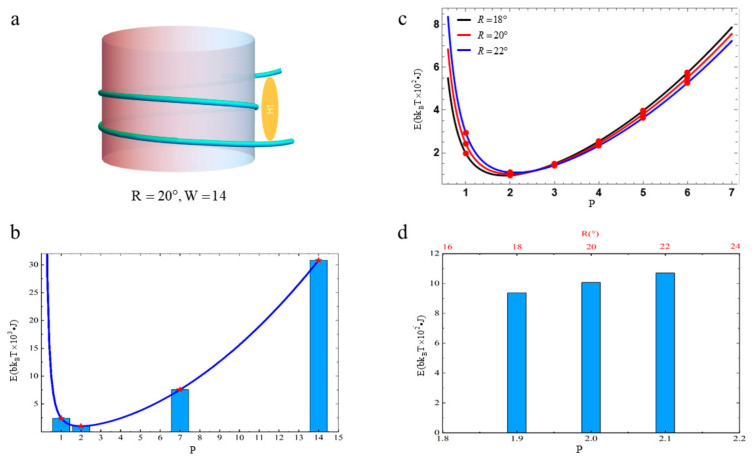
(**a**) Sketch of a typical nucleosome. (**b**) When m=n=1 and R=20°, the free energy profile of the nucleosomal DNA with P and the histogram of the free energy of the nucleosomal DNA with P=1,2,7,14 are plotted. (**c**) The free energy profiles are plotted with different R values. The red circles refer to the free energy’s values when P takes an integer. (**d**) The histogram shows the minimum of the free energy with different R values. The schematic diagram also devotes the values of P corresponding to the minimum free energy.

**Figure 6 biomolecules-13-01686-f006:**
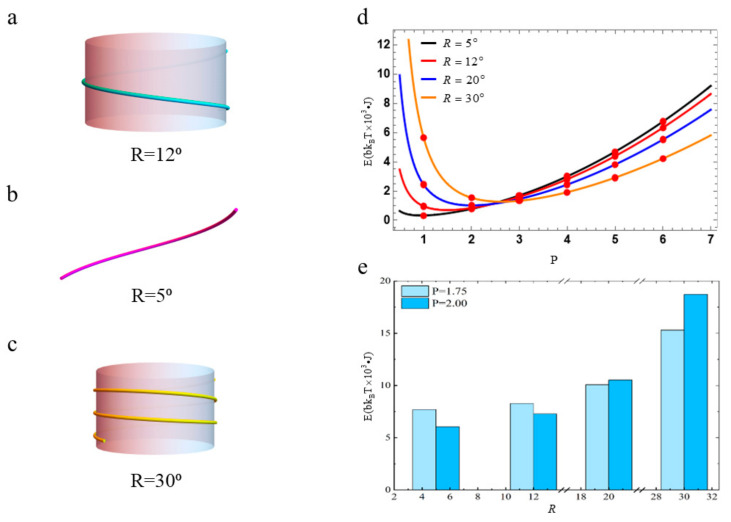
(**a**) Sketch of the NCPs when the eight histones are loosely bound. (**b**) Sketch of the NCPs when the eight histones are disassembled. (**c**) Sketch of the NCPs when the eight histones are more tightly bound. (**d**) The free energy profiles with R=5°,12°,20°,30°. The red circles represent the free energy’s values when P takes an integer. (**e**) Histogram of the free energy of DNA corresponding to the different angles at P=2, P=1.75 when W=14, m=1, n=1.

**Figure 7 biomolecules-13-01686-f007:**
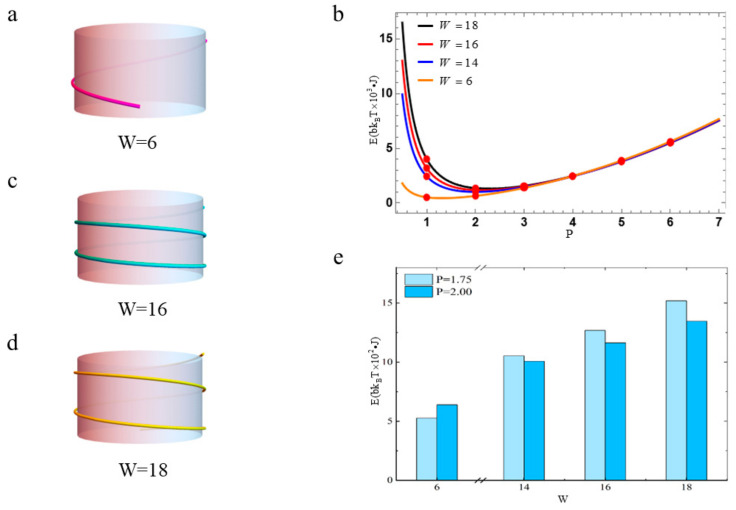
(**a**) Sketch of the NCPs for $L_{k}=6$. (**b**) The free energy profiles with $L_{k}=6,14,16,18$. The red circles represent the free energy’s values when P takes integer. (**c**) Sketch of the NCPs for $L_{k}=16$. (**d**) Sketch of the NCPs for $L_{k}=18$. (**e**) Histogram of the free energy of DNA corresponding to different $L_{k}$ values when R=20°, m=1, n=1, P=2, P=1.75.

## Data Availability

Data sharing is not applicable to this article as no new data were created or analyzed in this study.

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
