# Peer review of "The Free Energy of Nucleosomal DNA Based on the Landau Model and Topology"

_biomolecules, 2023, doi:10.3390/biom13121686_

Round 1
Reviewer 1 Report
Comments and Suggestions for Authors
I have reviewed the publication titled "Free Energy of Nucleosomal DNA Based on Landau Model and Topology" by [authors]. The aim of this paper is to explore the relationship between the free energy of nucleosomal DNA and its conformation, particularly topology. The authors have made significant contributions to the field by demonstrating that the free energy for stable nucleosomal DNA follows the energy minimum principle and that small perturbations have little effect on its free energy. The strengths of this paper include its clear and concise presentation of the research findings, as well as the use of a well-established theoretical framework to analyze the data.
In terms of general concept comments, I would like to highlight the need for further clarification on the experimental methods used to obtain the data. While the authors have provided a detailed description of the Landau model and topology, there is limited information on the specific experimental techniques used to measure the free energy of nucleosomal DNA. Additionally, it would be helpful to include more information on the limitations of the study, such as the potential impact of environmental factors on the stability of nucleosomal DNA.
In terms of the article, I would like to comment on the completeness of the review topic covered. The authors have done an excellent job of summarizing the current state of knowledge on the free energy of nucleosomal DNA, as well as identifying gaps in knowledge that need to be addressed in future research. However, I would suggest that the authors include more references to recent studies on the topic, as some of the cited references are quite dated.
Overall, I believe that this publication makes a valuable contribution to the field of nucleosomal DNA research. The authors have presented a well-reasoned and well-supported argument for the relationship between free energy and topology, and have identified important areas for future research. I would recommend this paper for publication with minor revisions.
Reviewer 2 Report
Comments and Suggestions for Authors
The work examines DNA within nucleosomes from a topological point of view. The influence of the geometric characteristics of the nucleosome core particle is considered.
1. Lines 34-47. In recent years, many works have been done about the nucleosome DNA’s conformation and free energy [6-10]. etc.
In [6-13] studies of DNA on histones at different levels of detail are presented. I think the atomistic level also would be interesting to mention (see, for example, https://doi.org/10.1371/journal.pcbi.1006224).
2. Lines 61-71.
It is worth expanding the description, adding White’s formula in the original notation.
3. Lines 84-85. ni is the number of base pairs per helix of DNA.
This probably means ni=10.5 base pairs per turn (not helix) of DNA.
4. Line 108 where its torsional persistence length of 95 t l = nm .
Why was this value of torsion persistent length chosen? Link missing.
5. Line 114 single-molecule FRET
What is FRET? Please, define the acronym when it first appears in text.
6. Line 153 clockwisea round -> clockwise around
Line 164. Linear DNA is the subject in many vitro experiments. -> in vitro
Correct the typos please.
7. Please add a description of the symbols A, B, C and angles in the caption to Figure 4.
8. The captions for Figure 5 are difficult to distinguish. In addition, the legend is poor in describing what is seen in the drawings. For example, what do the red circles in Figure 5b mean? It is not clear why not integer values of P are considered in Figure 5c, if P should take integer values (line 215)? In addition, just three points can say little about the nature of the curve. There is no explanation in the text why only the case n=m=1 is considered.
The sequence of references in the text to Figure 5 is also confusing. The text first refers to Figure 5c, then 5d, then 5a and 5b. This part of the article should be reformulated or the figures should be swapped. The same goes for Figure 6.
9. Lines 267-268. That means small changes in R have little effect on the nucleosomal DNA free energy. These show the conformation of DNA has some redundancy, that is, the small change in R affects the stability of DNA slightly.
This conclusion is not consistent with the previous text, where the difference was calculated for adjacent P values. Moreover, from a biological point of view, not all R values can be considered to belong to the DNA on a nucleosome.
10. Lines 386-387. 25. Landau, L. D.; Lifšic, E. M.; Lifshitz, E. M.; Kosevich, A. M.; Pitaevskii, L. P., Theory of elasticity: volume 7. Elsevier: 1986; 386 Vol. 7. 387
Lifšic, E. M. and Lifshitz, E. M. is the same author. Please, correct the reference.
Consideration of the problem is certainly of interest. The conclusions contain philosophical formulations. However, the results are more about the DNA itself and not about the protein. Do you think that without histone proteins, DNA would fold in the same way? But why doesn't this happen in a cell? Is it possible to consider supercoiled DNA using the described method?
In addition, histones are not the only proteins on which DNA bending occurs. I would like to hear a forecast about the applicability of the method on other proteins, for example, bacterial Fis, HU and others.
Reviewer 3 Report
Comments and Suggestions for Authors
Please, see attached.

There typos like
Line 48: a typo “It is interesting to know what the free energy of nucleosomal DNA behaves?” consider “how the free energy […] behaves”
or instead "formula/formulas" there appears "formular", perhaps "equation" should be more appropriate
Then there are sentences hard to understand like "Generally speaking, such line macromolecular are easy to tangle and cause confusion."
etc.
Round 2
Reviewer 3 Report
Comments and Suggestions for Authors
I have read the revised manuscript by Huimin Yang et al entitled “The free energy of nucleosomal DNA based on Landau model and topology”. The Authors revised the manuscript extensively and responded to all my comments. As such the request for major revisions was satisfied.
I would like to have a few more questions.
1) The parametrization of the model by Okushima, T. et al (PHYSICAL REVIEW E 84, 021926 (2011)) simulating the experimental setup of Lee, M. et al (Proc. Natl. Acad. Sci. U.S. A. 2010, 107 (11), 4985-90) is done for a special sequence of DNA consisting from guanine-cytosine DNA base-pairs, which were chosen because they are prone to undergo transition to Z-form upon twisting. However, the DNA is rather a co-polymer of AT and GC base-pairs than a poly(GC) polymer. Can the Authors estimate and comment how robust is their model and if the minima of the free energy would be still predicted at 1.75-2 turns of DNA around nucleosome if the composition of the DNA changed?
2) Authors used White’s formula to estimate the value of the linking number as a sum of intrinsic DNA twisting and writhing induced by wrapping of DNA around the nucleosome proteins. The theoretical excess of linking number originating from wrapping of the DNA should be around -2, since delta_Lk = delta_Tw + delta_Wr (if the twist is unchanged by interactions with the proteins, delta_Tw = 0). In the current setting of the model, the authors take value of 1.75 turns of DNA around a nucleosome, hence, the excess of the linking they consider is delta_Lk = - 1.75, in line with the theoretical expectations. But, the experimental measurements revealed, that the excess linking number is smaller, delta_Lk = -1.26 (Segura et al, Nature Communications volume 9, Article number: 3989 (2018)). The difference may not look to be very big. But, imagine, that one has a long circular chromosome of the yeast, which is a eukaryotic cell, having its DNA wrapped around tens of thousands of nucleosomes. Now, in experiments, it is posible to dissolve the nucleosomes, so one stays with the naked DNA. Because the chromosomal DNA of the yeast is circular, the DNA cannot relax the torsional stress after dissolving of nucleosomes, and the writhing that was formerly contained in nucleosomes is conserved and translates into DNA supercoiling. The supercoiling is lower than it would be theoretically expected. This is known as the linking number paradox on nucleosomes and it has been a long-standing mystery. My question, of course, is not to ask the authors to resolve this mystery. But, I’m curious if the model would produce different minima of the free energy, if the changes of the intrinsic twisting would be taken into consideration. Can Authors comment on this?
3) In line 35-36, the Authors say that “The linker DNA is locked at the entry and exit points of NCPs by histone H1.” But, does H1 prevent also axial rotations of DNA?
Since the journal requires also evaluation of English that needs to be attached to my comments, I would say, the English was somewhat improved, but there is still a substantial amount of typos, that need to be corrected, for example:
Line 22: typo macromolecule, not "macromolecular"
Line 25: have evolved nucleosomes, which wrap the DNA around instead of "which DNA wraps around"
Line 29: various biological functions, typo in the word function that should be plural
Comments on the Quality of English LanguageEnglish was somewhat improved, but there is still a substantial amount of typos, that need to be corrected, for example:
Line 22: typo macromolecule, not "macromolecular"
Line 25: have evolved nucleosomes, which wrap the DNA around instead of "which DNA wraps around"
Line 29: various biological functions, typo in the word function that should be plural
